# The Association between Gestational Age and Risk for Long Term Ophthalmic Morbidities among Offspring Delivered in Different Preterm Subgroups

**DOI:** 10.3390/jcm11092562

**Published:** 2022-05-02

**Authors:** Erez Tsumi, Itai Hazan, Tamir Regev, Samuel Leeman, Chiya Barrett, Noa Fried Regev, Eyal Sheiner

**Affiliations:** 1Department of Ophthalmology, Soroka University Medical Center, Ben-Gurion University of the Negev, Beer-Sheva 8410101, Israel; tamirre@clalit.org.il (T.R.); leemans@post.bgu.ac.il (S.L.); barat@post.bgu.ac.il (C.B.); 2Joyce and Irving Goldman Medical School, Faculty of Health Sciences, Ben-Gurion University of the Negev, Beer-Sheva 8410501, Israel; itaihaz@post.bgu.ac.il; 3Department of Emergency Medicine, Soroka University Medical Center, Ben-Gurion University of the Negev, Beer-Sheva 8410101, Israel; noafri@clalit.org.il; 4Department of Obstetrics and Gynecology, Soroka University Medical Center, Ben-Gurion University of the Negev, Beer-Sheva 8410101, Israel

**Keywords:** ophthalmic morbidities, retinopathy of prematurity, gestational age, preterm delivery

## Abstract

Objective: To investigate whether there is a linear association between the degree of prematurity and the risk for long-term ophthalmic morbidity among preterm infants. Study design: A population-based, retrospective cohort study, which included all singleton deliveries occurring between 1991 and 2014 at a single tertiary medical center. All infants were divided into four groups according to gestational age categories: extremely preterm births, very preterm births, moderate to late preterm births and term deliveries (reference group). Hospitalizations of offspring up to 18 years of age involving ophthalmic morbidity were evaluated. Survival curves compared cumulative hospitalizations and regression models controlled for confounding variables. Results: During the study period, 243,363 deliveries met the inclusion criteria. Ophthalmic-related hospitalization rates were lower among children born at term (0.9%) as compared with extremely preterm (3.6%), very preterm (2%), and moderate to late preterm (1.4%) born offspring (*p* < 0.01; using the chi-square test for trends). The survival curve demonstrated significantly different hospitalization rates between the gestational ages (*p* < 0.001). The regression demonstrated an independent risk for ophthalmic morbidity among extremely preterm born offspring (adjusted hazard ratio 3.8, confidence interval 1.6–9.2, *p* < 0.01), as well as very preterm and moderate to late preterm (adjusted hazard ratio 2.2 and 1.5, respectively) as compared with term deliveries. Conclusions: The risk for long-term ophthalmic-related hospitalization of preterm offspring gradually decreases as the gestational age increases.

## 1. Introduction

The consequences of prematurity are well established in literature, causing long-term and potentially severe effects on pediatric and infantile morbidity and mortality [1,2]. With incomplete maturation, preterm infants are at a greater risk for developing a wide spectrum of medical complications, including hypothermia, jaundice, respiratory disorders [3], immunologic problems [4], neurodevelopmental problems [5,6,7], increased susceptibility to infections, hypoglycemia and feeding problems. In accordance with preterm newborns’ shorter intrauterine periods, ocular development in preterm infants is also subject to a number of abnormal influences, given the reduced time for proper growth and support in a unique environment [8]. Previous literature demonstrated that children who were born very preterm (28–32 weeks) are at a significantly higher risk for abnormal visual and neurological development, when compared to children born at full term [9]. These abnormalities include retinopathy of prematurity (ROP), strabismus, color vision deficits, visual field defects, decreased visual acuity and refractive error [10].

The World Health Organization (WHO) subcategorizes preterm births based on gestational age: extremely preterm (less than 28 weeks delivery), very preterm (28 to 32 weeks delivery), moderate to late preterm (32 to 37 weeks delivery) [11].

A recent study showed that long-term ophthalmic morbidities of offspring is significantly associated with early term delivery [12]. Early preterm-born offspring were found to have an independent risk for long-term ophthalmic morbidity (adjusted hazard ratio 2.51, confidence interval 1.91–3.29) as compared with full term offspring. In our study, we sought to investigate one step further to understand the relative risk for long-term ophthalmic morbidity between the different subcategories of preterm deliveries.

## 2. Materials and Methods

A retrospective cohort study of all singleton pregnancies in women who gave birth between the years 1991 and 2014 was conducted. Data was taken from the Soroka Univer-sity Medical Center (SUMC), the major tertiary hospital in the Negev region of Israel and the largest birth center in the country. The Negev region has continued to see increasing immigration since the 1990s. Therefore, the present study was based on non-selective population data. The study protocol was a received informed consent exemption by the SUMC institutional review board and was exempt from informed consent. The study population included two different ethnic groups: Jewish and Bedouins, who differ in their economic status, levels of education, and traditional beliefs [13]. However, prenatal care services are available to all Israeli citizens free of charge (covered by universal national health Insurance) [13]. Nevertheless, prenatal care services utilization is lower in Bedouin women as compared to Jewish women for a variety of social, cultural, and geographical access issues [14,15].

The primary exposure was defined as pre-term delivery (before 37 weeks). The control groups consisted of newborns born at later gestations. Gestational age was based on the best obstetric estimate determined by providers and used for clinical decision making. The standard criteria used involved consideration of the clinical history and earliest ultrasound scan. If the last menstrual period (LMP) was certain and consistent with the ultrasound, dating was based on the LMP. If the ultrasound was not consistent with the LMP, or the LMP was unknown, ultrasound data were used to determine gestational age.

Excluded from the study were fetuses with congenital malformations or chromosomal abnormalities, as well as perinatal mortality cases (intrauterine fetal death, intra-partum death, and post-partum death) and nonsingleton births. All offspring were divided into four groups according to their gestational age at delivery: extremely preterm birth: 24–28 gestational weeks, very preterm birth: 28–32 gestational weeks, moderate to late preterm birth: 32–37 gestational weeks and term deliveries: above 37 weeks.

The long-term outcomes assessed included all hospitalizations of offspring at SUMC until the age of 18 years that involved ophthalmic morbidity. The term “ophthalmic morbidity” included four categories: visual disturbances, retinopathy of prematurity, ocular infection and hospitalization. All diagnoses during hospitalization were predefined according to a set of ICD-9 procedures and diagnostic codes, detailed in Table A1.

Follow-up was terminated once any of the following occurred: after the first hospitalization, due to any of the predefined ophthalmic morbidities, any hospitalization resulting in death of the child, or when the child reached 18 years of age (calculated by date of birth).

Data was collected from two databases that were cross-linked and merged: the computerized hospitalization database of SUMC (“Demog-ICD9”), and the computerized perinatal database of the Obstetrics and Gynecology Department. The perinatal database consists of information recorded immediately following an obstetrician delivery. Experienced medical secretaries routinely reviewed the information prior to entering it into the database to ensure maximal completeness and accuracy. Coding was carried out after assessing medical prenatal care records, as well as routine hospital documents.

The SPSS package 23rd ed. (IBM/SPSS, Chicago, IL, USA) was used for statistical analysis. Categorical data is shown in counts and rates. Associations between the gestational age categories, background and outcome characteristics were assessed using the chi-square and ANOVA tests. In order to demonstrate the cumulative hospitalization incidences over time among the study groups, a Kaplan–Meier survival curve was used, and the log-rank test was used to assess the difference between the curves.

For the purposes of establishing an independent association between specific gestational age and the future incidence of ophthalmic-related hospitalizations of the offspring, a Cox regression model was constructed. The model was adjusted for confounding and clinically significant variables, including maternal age and diabetes (pre-gestational and gestational). All analyses were two-sided, and a *p*-value of ≤0.05 was statistically considered.

## 3. Results

A total of 243,363 deliveries were included in the study; 405 were between 24–28 gestational weeks (0.2%), 1084 deliveries were between 28–32 weeks (0.4%), 14,956 deliveries were between 32–37 weeks (6.1%) and 226,918 occurred at ≥37 gestational weeks (93.2%).

Table 1 summarizes maternal characteristics by delivery week category. Maternal age was similar in all delivery groups, with an overall average of 28.08 ± 6.3. Diabetes was more likely to occur in deliveries between 32–37 weeks (7.5%). Hypertensive disorders were more common in deliveries between 28–32 weeks (17.7%).

Table 2 summarizes pregnancy outcomes of all four groups. Mean birth weight was positively associated to gestational age; ranging from 855.1 ± 429.5 for 24–28 weeks of age to 3264 ± 446.6 among the mature group, while small for gestational age (SGA) as well as low birth weight (LBW( were most common in the former group (13.3% and 97.5% respectively). Cesarean deliveries were significantly more common among women in 28–32 weeks of gestation (44.8%, *p*-value < 0.01).

The long-term ophthalmological morbidities based on hospitalizations is presented in Table 3. Rates of visual disturbances and retinopathy of prematurity (ROP) were significantly higher among the 24–28 gestational weeks group (0.7%, *p*-value < 0.01; 1.4%, *p*-value < 0.01 respectively), while ocular infection was highest among newborns in the 28–32 weeks of gestation (0.9%, *p*-value < 0.01). Total ophthalmic hospitalization rates were highest among the 24–28 gestational weeks group (3.6%, *p*-value < 0.01 using the chi-square test for trends), and gradually decreased with increasing gestational week (2% among 28–32 weeks, 1.4% among 32–37 weeks, and 0.9% among 37+ weeks). A decrease in the number of hospitalizations with gestational age was also documented while stratifying for ethnicity. Offspring born between 24–28 weeks had the highest cumulative incidence of hospitalizations (Figure 1) due to ophthalmological morbidity, followed by offspring born at 28-36 and those born at ≥37 gestational weeks (*p* < 0.001).

In the multivariate Cox regression model for offspring long-term risk of ophthalmic-related hospitalizations (Table 4), earlier deliveries were associated with higher risk for hospitalization, compared to the most mature group (24–28) weeks: HR = 3.8; CI95% 2.6–9.2; 28–32 weeks: HR = 2.2; CI95% 1.4–3.5; 32–37 weeks: HR = 1.5; CI95% 1.3–1.7).

## 4. Discussion

This large population-based cohort study demonstrates an increased risk for childhood and adolescence long-term ophthalmic morbidities as the gestational age becomes earlier among the pre-term deliveries. The risk remained significantly elevated among all the preterm groups while controlling for relevant maternal factors.

Numerous studies have identified an association between early gestational age at birth and the risk of poor health outcomes, based on incidence of morbidities, mortality and hospitalization rates [16,17]. A growing body of evidence has also shown ophthalmic morbidity among preterm infants. Kozeis et al. [18] found preterm infants to be associated with impairment of some aspects of visual function.

Although the pathophysiology is still not fully understood, complex multifactorial mechanisms have been suggested as the potential causes for ocular damage from preterm delivery. The suggested mechanisms include deficiencies in both innate and adaptive immunity [19], hypoxic-ischemic induced inflammation and cytokine injury, reperfusion injury, toxin-mediated injury, infection [20,21], and insufficient endogenous hormones (e.g., Cortisol & thyroxin), often exhibited as transient hypothyroxinemia of prematurity (THOP) [22].

Vast epidemiological studies conducted worldwide on ROP showed that despite geographical variability, the incidence of ROP was similar [14,15,16,17,18]. A study conducted by the Australian and New Zealand Neonatal Network (ANZNN) showed that the incidence of severe ROP was higher (34%) in infants born before 25 weeks gestation when compared to infants born at 25–26 weeks gestation (12.9%) [23]. Similar results have been found in North America, the UK and Indonesia [24,25,26].

Our findings revealed that the highest rate of ROP was observed in infants born at 24–28 weeks gestation (1.4%), with a declining rate for more advanced gestational age and reaching zero cases in those born after 32 weeks’ gestation. These results are consistent with previous studies examining the rate of ROP [27,28,29,30] and are likely to be explained by the fact that ROP is a disease of developing blood vessels, which are still considered underdeveloped at the beginning of the third trimester [31].

Additional long-term ophthalmological morbidities, such as visual disturbances, were examined in our study and showed the same pattern. The higher visual disturbances in preterm babies are a consequence of the higher retinopathy rate and other reasons e.g., osteopenia of prematurity [32]. Other types of vision disturbances include sub-categories, such as strabismus (both exotropia and esotropia), refractive errors, visual field deficits, color vision errors, astigmatism, cortical blindness and more. A recent study comparing strabismus in premature and term children found that the risk for strabismus in premature children was substantially higher than in full-term children (16.2% vs. 3.2%) [33]. Other ophthalmological morbidities related to prematurity, such as ROP, were linked to increased risk for strabismus as well [34].

Many factors contribute to the increased susceptibility of preterm infants to infections. Prematurity is associated with underdeveloped ocular surface defense mechanisms and has also been associated with neonatal conjunctivitis [35]. While previous studies showed an increase in the risk for general ocular infection among preterm newborns compared to term newborns [36], our analysis did not show a conclusive trend. Infants born between 24–28 weeks of gestation showed a lower proportion of ocular infections compared to those born during weeks 28–32 (0.7% compared to 0.9% respectively). This result could be explained by the smaller size of the preterm group and requires further analysis.

The key strength of our study was the use of a large cohort in a single medical center providing broad follow-up of children up to 18 years of age. This allowed for the investigation of substantially larger numbers of infants to be followed-up on than could be achieved by individual NICU follow-up studies, and simplified the monitoring of all infants for the purposes of comparison.

A major limitation of our study is its retrospective nature, as such studies may only suggest an association rather than causation. An additional limitation is the fact that information on other important risk factors, such as lifestyle, nutrition, perinatal treatment and other family factors was not available in the datasets. Two more challenges stemmed from the choice of ophthalmic-related hospitalizations as the present study’s endpoint: (1) Most of the ophthalmic morbidity is probably catered to in an outpatient setting (as discussed earlier). However, all significant ocular morbidity is routed to the SUMC, as it is the only tertiary hospital in the South of Israel (2) Offspring born earlier are more likely to be hospitalized in general, and therefore may be more frequently diagnosed with ophthalmic morbidities.

## 5. Conclusions

Our study was able to find a significant association between degree of prematurity and long-term ocular morbidities up to 18 years of age. A markedly increasing risk of severe adverse neonatal outcomes was observed as gestational age decreased. Further studies could raise awareness for early interventions, improve children’s health outcomes and decrease the burden on health care systems attributed to these diseases.

## Figures and Tables

**Figure 1 jcm-11-02562-f001:**
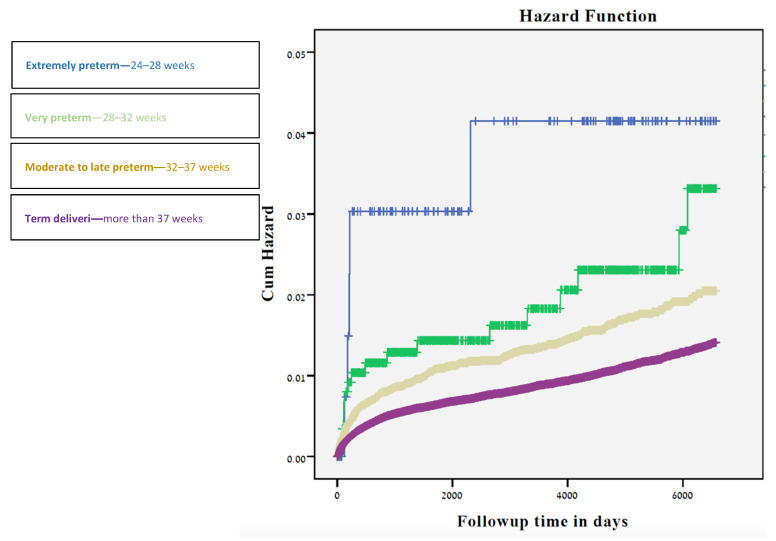
Kaplan-Meier survival curve demonstrating the cumulative incidence of. ophthalmological-related hospitalizations compared to delivery time in weeks.

**Table 1 jcm-11-02562-t001:** Maternal characteristics according to gestational age.

Maternal Characteristic	Extremely Preterm: 24–28 Weeks (*n* = 405)	Very Preterm: 28–32 Weeks (*n* = 1084)	Moderate to Late Preterm: 32–37 Weeks (*n* = 14,956)	Term Deliveries: more than 37 weeks (*n* = 22,6918)	*p*-Value
Ethnicity, *n* (%)					<0.01
Jewish	190 (46.9%)	463 (42.7%)	6883 (46%)	107,673 (47.5%)	
Bedouin	215 (53.1%)	621 (57.3%)	8073 (54%)	119,245 (52.5%)	
Maternal age, years, mean ± SD	28.44 ± 6.5	28.16 ± 6.3	28.14 ± 6.2	28.16 ± 5.7	<0.01
Diabetes ^1^, *n* (%)	5 (1.2%)	57 (5.3%)	1118 (7.5%)	10,973 (4.8%)	<0.01
Hypertensive disease ^2^, *n* (%)	40 (9.9%)	192 (17.7%)	1839 (12.3%)	10,169 (4.5%)	<0.01

^1^ Including pre gestational and gestational diabetes; ^2^ Including pre gestational, gestational hypertension and pre- eclampsia.

**Table 2 jcm-11-02562-t002:** Pregnancy outcomes for children (age < 18) by delivery week.

Pregnancy Outcome	Extremely Preterm: 24–28 Weeks (*n* = 405)	Very Preterm: 28–32 Weeks (*n* = 1084)	Moderate to Late Preterm: 32–37 Weeks (*n* = 14,676)	Term Deliveries: more than 37 Weeks (*n* = 226,917)	*p*-Value
Birthweight, gr mean ± SD	855.1 ± 429.5	1560.7 ± 623.4	2532.4 ± 506.3	3264.2 ± 446.6	<0.01
Small for gestational age ^1^, *n* (%)	54 (13.3%)	54 (5%)	607 (4.1%)	10,547 (4.6%)	<0.01
Low birth weight ^2^, *n* (%)	395 (97.5%)	992 (91.5%)	7030 (47%)	7805 (3.4%)	<0.01
Cesarean delivery, *n* (%)	139 (34.3%)	486 (44.8%)	4444 (29.7%)	27,908 (12.3%)	<0.01

^1^ Small for gestational age (SGA) < 5th percentile for gestational age; ^2^ Low birth weight (LBW) < 2500 g.

**Table 3 jcm-11-02562-t003:** Selected long-term ophthalmological morbidities for children (age < 18) by delivery week.

Ophthalmological Morbidity		Extremely Preterm: 24–28 Weeks (*n* = 138)	Very Preterm: 28–32 Weeks (*n* = 891)	Moderate to Late Preterm: 32–37 Weeks (*n* = 14,676)	Term Deliveries: more than 37 Weeks (*n* = 226,482)	*p*-Value
Visual disturbances, *n* (%)		1 (0.7%)	0 (0%)	36 (0.2%)	245 (0.1%)	<0.01
ROP, *n* (%)		2 (1.4%)	2 (0.2%)	0 (0%)	0 (0%)	<0.01
Ocular Infections, *n* (%)		1 (0.7%)	8 (0.9%)	123 (0.8%)	1336 (0.6%)	0.02
Total ophthalmic Hospitalization, *n* (%)	Bedouin	2 (1.4%)	12 (1.3%)	123 (0.8%)	1190 (0.5%)	<0.01
	Jewish	3 (2.2%)	6 (0.7%)	85 (0.6%)	907 (0.4%)	<0.01
	All	5 (3.6%)	18 (2%)	208 (1.4%)	2097 (0.9%)	<0.01

**Table 4 jcm-11-02562-t004:** Cox regression analysis of the association between week of birth and ophthalmic-related hospitalization of children (age < 18).

	Hazard Ratio	95% CI	*p*-Value
Extremely preterm: 24–28 weeks	3.8	1.6–9.2	<0.01
Very preterm: 28–32 weeks	2.2	1.4–3.5	<0.01
Moderate to late preterm: 32–37 weeks	1.5	1.3–1.7	<0.01
Term deliveries: more than 37 weeks	1 (Reference)		
Mother age at birth	0.9	0.98–0.99	0.02
Diabetes	1.0	0.8–1.2	0.6

## Data Availability

Unavailable according to the Helsinki protocol.

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
