# Peer review of "The Association between Gestational Age and Risk for Long Term Ophthalmic Morbidities among Offspring Delivered in Different Preterm Subgroups"

_jcm, 2022, doi:10.3390/jcm11092562_

Round 1

Reviewer 1 Report

This study confirmed what is already known, the lower the gestational age at birth, the higher the incidence of retinopathy. Higher visual disturbances in the most preterm babies are a consequence of the higher retinopathy and other reasons e.g. osteopenia of prematurity (see Pohlandt, 1994, Eur J Pediatr, Vol 153, p234-236) unfortunately not addressed here. The results for ocular infections show no clear trend affected by prematurity. 

It is not clear, what the p-values in the tables stand for.

Reviewer 2 Report

The study has two different populations Jewish and Bedouin, they completely differ in their socio economic state and the community services (the medical including the pre natal care as well) that are available to them. Therefore all results should be presented for each group separately. I presume the results differ between the two groups. This should be presented and discussed.

I doubt whether the results of ophthalmic morbidity should be presented since most morbidities are treated in ophthalmological practices and only a minority will be sent to the hospital.
The data about ophthalmic morbidity are definitely inaccurate, they certainly underestimate the morbidity.

The main problem, as the authors point out, the data about visual disturbances is much lower than it is in reality. The data is from the database of SUMC but most ophthalmological problems are treated in the community, and certainly in Israel.

It is questionable whether those data should be presented.

On the other hand the hospitalisation rate is definitely very interesting, especially how the two groups differ.
